# Epithelial polarization in the 3D matrix requires MST3 signaling to regulate ZO-1 position

Chee-Hong Chan[1☯], Pei Lin[2☯], Tse-Yen Yang[3], Bo-Ying Bao[4], Jhen-Yang Jhong[5], Yui-Ping Weng[6], Te-Hsiu Lee[7], Hui-Fen Cheng[8], Te-Ling Lu[4]*

1 Department of Nephrology, Chang Bing Show Chwan Memorial Hospital, Lukang, Changhua, Taiwan, 2 Division of Cardiology, Department of Internal Medicine, An Nan Hospital, China Medical University, Tainan, Taiwan, 3 Molecular and Genomic Epidemiology Center, Department of Medical Research, China Medical University, Tainan, Taiwan, 4 College of School of Pharmacy, China Medical University, Tainan, Taiwan, 5 Department of Medical Laboratory Science and Biotechnology, Sin-Lau Hospital, Tainan, Taiwan, 6 Department of Acupressure Technology, Chung Hwa University of Medical Technology, Tainan, Taiwan, 7 Department of Medical Laboratory Science and Biotechnology, Chung Hwa University of Medical Technology, Tainan, Taiwan, 8 Department of Laboratory Medicine, Tainan Municipal Hospital (Managed by Show Chwan Medical Care Corporation), Tainan, Taiwan

☯ These authors contributed equally to this work.
* lutl@mail.cmu.edu.tw

**Data Availability Statement:** All relevant data are within the paper and its Supporting Information files.

## Abstract

Apical-basal cell polarity must be tightly controlled for epithelial cyst and tubule formation, and these are important functional units in various epithelial organs. Polarization is achieved through the coordination of several molecules that divide cells into an apical domain and a basolateral domain, which are separated from tight and adherens junctions. Cdc42 regulates cytoskeletal organization and the tight junction protein ZO-1 at the apical margin of epithelial cell junctions. MST kinases control organ size through the regulation of cell proliferation and cell polarity. For example, MST1 relays the Rap1 signal to induce cell polarity and adhesion of lymphocytes. Our previous study showed that MST3 was involved in E-cadherin regulation and migration in MCF7 cells. *In vivo*, MST3 knockout mice exhibited higher ENaC expression at the apical site of renal tubules, resulting in hypertension. However, it was not clear whether MST3 was involved in cell polarity. Here, control MDCK cells, HA-MST3 and HA-MST3 kinase-dead (HA-MST3-KD) overexpressing MDCK cells were cultured in collagen or Matrigel. We found that the cysts of HA-MST3 cells were fewer and smaller than those of control MDCK cells; ZO-1 was delayed to the apical site of cysts and in cell-cell contact in the $Ca^{2+}$ switch assay. However, HA-MST3-KD cells exhibited multilumen cysts. Intensive F-actin stress fibers were observed in HA-MST3 cells with higher Cdc42 activity; in contrast, HA-MST3-KD cells had lower Cdc42 activity and weaker F-actin staining. In this study, we identified a new MST3 function in the establishment of cell polarity through Cdc42 regulation.

**Funding:** This work was supported by grants from the National Science Council of Taiwan research MOST 109-2320-B-039-017, China Medical University Grant CMU110-MF-76, Chang Bing Show-Chwan Memorial Hospital Grant BRD109001, and An Nan Hospital-China Medical University in Taiwan Grant ANHRF111-10. The funders had no role in study design, data collection and analysis, decision to publish, or preparation of the manuscript.

**Competing interests:** The authors have declared that no competing interests exist.

## Introduction

Formation of the lumen is a fundamental step in epithelial organ development. The kidneys are among the best-characterized organs in the lumen and tubes of renal development. When MDCK cells are embedded into the extracellular matrix (ECM), the cells interpret signals from the ECM and transduce them to form cysts with a hollow lumen. The process requires regulators to mediate a balance between proliferation, apoptosis and generation of the axis of polarity [1].

For full polarization, the adheren junctional proteins (AJs), E-cadherin (E-cad), form an adhesive belt that encircles each epithelial cell just underneath the apical surface. ZO-1 is localized at primordial AJs in the initial phase of epithelial polarization [2, 3] and eventually localizes at tight junctions (TJs) after the maturation of epithelial polarization [4, 5]. Cdc42 is a member of the family of Ras-like GTPases. The assembly of Cdc42 and Cdc42-associated proteins is important for the formation of TJs. ZO-1 physically interacts with Cdc42, which is involved in cell-cell contact and the membrane protrusions of motile cells [6]. At the beginning of cyst formation (Day 1), Cdc42 localized at cell-cell and cell-ECM contacts; thereafter (Days 2–5), the majority of Cdc42 localized at apical contacts, delimiting the newly formed lumen [7]. Both constitutively active and dominant-negative Cdc42 prevent cytoskeletal polarization. Constitutively activated Cdc42 disrupts tight junctions by slowing endocytic and biosynthetic traffic in MDCK epithelial cells [8]. In 3D culture, constitutively activated Cdc42 induces ZO-1 translocation at the basal site instead of the apical site in MDCK cells, indicating that constitutively activated Cdc42 affects ZO-1 basal redistribution and loss of cell polarity [9]. When Cdc42 function is downregulated by siRNA knockdown, MDCK cells form multiple small lumens in most cysts [10]. These results indicate that Cdc42 plays a role in sorting associated proteins into distinct apical or basolateral vesicles for the final target membranes.

MST kinases (mammalian sterile 20-like kinases) are composed of MST1, MST2, MST3, MST4 and YSK1, sharing a conserved kinase domain in the N-terminal region with diverse C-terminal regions that mediate protein-protein interactions. They play crucial roles in cell polarity through the regulation of different GTPases in different cells [11]. MST1 regulates lymphocyte polarization and adhesion through Rap1 and RAPL [12]. MST1 and MST2 kinases also regulate thymocyte migration through Rac1 activation [13]. Deletion of Cdc42 has been shown to suppress MST1 and MST2 phosphorylation, leading to intestinal cell hyperplasia and defective polarity [14]. MST4 (also named STK26) regulates brush border formation in colorectal adenocarcinoma cells, but the polarized localization of ZO-1 is not altered [15].

Our previous results showed that MST3 (also named STK24) is proteolytically cleaved by caspase and that cleaved MST3 results in kinase activation, nuclear translocation and the induction of apoptosis [16]. In addition, we also showed that overexpression of MST3 inhibited migration and spread in MDCK cells. In contrast, suppression of MST3 by siRNA enhanced cellular migration in MCF7 cells with reduced expression of E-cad at the edge of migrating cells [17]. In an *in vivo* study, MST3 knockout mice exhibited higher ENaC apical expression in renal tubules [18]. We also found that spontaneous hypertensive mice with higher ENaC expression had lower MST3 expression than control mice [19]. ENaC is a sodium channel present at the apical site of renal collecting ducts that is also regulated through Ras GTPase-mediated trafficking [20]; therefore, we hypothesize that MST3 plays a role in protein trafficking, leading to cell polarization. To determine whether MST3 is involved in cell polarization, we evaluated F-actin structures, Cdc42 activity and polarity of cysts in overexpressed HA-tagged wild type (WT)- and kinase dead (KD)-MST3 MDCK cells. We provide evidence that MST3 controls MDCK cyst formation through apoptosis and Cdc42 regulation.

## Materials and methods

### Stable clones and 3D culture

MDCK cells were obtained from the Bioresource Collection and Research Center (BCRC, Taiwan) and maintained in DMEM containing 10% FBS and antibiotics at 37˚C with 5% $CO_2$ in a humidified incubator. Before transient transfection, approximately $8 \times 10^5$ cells were seeded in 6-well plates for one day. Approximately 1.6 μg of HA-MST3 or HA-MST3-K53R (HA-MST3-KD) in 125 μl OPTI medium was incubated with 20 μl LF2000 in 125 μl OPTI medium for 30 min and then added to the cells. MDCK cells were incubated in regular growth medium containing 800 μg/ml G418 to select stable transfectants. Matrigel (BD Biosciences, Cat. # 354230) was thawed overnight at 4˚C before use. Matrigel was diluted with DMEM to a concentration of 5 mg/mL on ice. Neutralized rat type I collagen (Corning, Cat. # 354236) solution containing DMEM (1X), $NaHCO_3$ (2.35 mg/ml), Hepes (20 mM), glutamax (24 mM), and collagen (2 mg/ml) was prepared. Diluted Matrigel or collagen was plated onto filters (Thermo Fisher Scientific, cat. No. 140652) that were incubated in a 37˚C, $CO_2$-free oven to allow collagen to polymerize into a gel. Cyst cultures were prepared as previously described [21]. Briefly, cells were trypsinized into a single-cell suspension of $1 \times 10^6$ cells/ml. Then, $7 \times 10^4$ cells in Matrigel or collagen solution were plated onto the cell-free gel layer. The Matrigel or collagen mixture containing cells was allowed to gel at 37˚C, and then medium was added. The medium was changed every 2 days until cysts with lumens formed.

### Phos-tag gel and western blot

MDCK cells were lysed with lysis buffer (50 mM Tris, pH 7.4, 150 mM NaCl and 1% Triton X-100). 50 μg cell lysates were incubated with 1 μg of recombinant protein phosphatase 2A C subunit (Cayman) at 30˚C for 30 min. The supernatants with 3X sample buffer (130 mM Tris, pH 6.8, 4% SDS, 10% 2-ME, 20% glycerol and 0.04% bromophenol blue) were denatured by heating at 95˚C for 5 min. The same amounts of protein were run on 8% separating SDS/PAGE gels or gel containing 50 μM Phos-tag and 100 μM $MnCl_2$. After proteins separated, the proteins in gel were transferred to nitrocellulose membranes. Membranes were incubated with primary antibodies and secondary antibodies (Table 1). The MST3 antibody against MST3, a kind gift from Dr. Ming-Derg Lai, was validated in our previous publications [17, 18]. Bands were detected by Western Lightning Plus (Perkin Elmer) using LAS2000. Representative images were uniformly processed in Adobe Photoshop.

### $Ca^{2+}$ switch assay

To remove $Ca^{2+}$ in FBS, FBS was dialyzed with Slide-A-Lyzer Dialysis Cassettes, 10K (Thermo Fisher Scientific, Cat. # 87732), at 4˚C against a buffer containing 150 mM NaCl and 50 mM $Na_2HPO_4$ pH 7.2 for 24 h, against the same buffer with 0.2 mM EDTA for 24 h, and then twice more against the same buffer without EDTA for 24 h. A total of $5 \times 10^5$ cells were plated onto 2 mg/ml collagen coated coverslips for 3 days. The cells were washed and incubated for 24 h in $Ca^{2+}$-free DMEM containing 10% dialyzed FBS and 15 mM HEPES. Then, the cells were replaced with standard DMEM containing $Ca^{2+}$ for 1 or 4 h.

### Immunofluorescence staining and confocal microscopy

The cysts in 3D culture were washed with PBS 3 times and incubated with 100 U/mL collagenase (Sigma, C0130) at 37˚C for 10 min. After washing with PBS, cells were fixed with 4% PFA at room temperature for 30 min. The cysts were permeabilized with PBS/10% BSA/0.5% Triton X-100 for 30 min. 2D cells were fixed with 4% PFA at room temperature for 10 min and

**Table 1. List of antibodies used for western blot and immunofluorescence staining.**

| Antigen | Species | Source | Dilution (IF) | Dilution (WB) |
|---|---|---|---|---|
| HA | Rat | Roche (clone 3F10, 12158167001) | 1:100 | 1:1000 |
| HA-Fluorescein | Rat | Roche (11988506001) | 1:100 | |
| MST3 | Rabbit | a gift from Dr. Lai. | N/A | 1:3000 |
| Cleaved Caspase3 | Rabbit | Cell Signallng (clone 5A1E, 9664) | 1:100 | 1:500 |
| Cdc42 | mouse | Cytoskeleton Inc. (Cat#ACD03) | N/A | 1:250 |
| Ki67 | Rabbit | Abcam (clone SP6, ab16667) | 1:100 | N/A |
| ZO-1 | Rabbit | Thermo Fisher Scientific (617300) | 1:400 | N/A |
| E-cadherin | mouse | BD Transduction Laboratories (610182) | 1:400 | N/A |
| b-actin | mouse | Sigma-Aldrich (clone AC-15, A1978) | N/A | 1:3000 |
| FITC- AffiniPure Goat Anti-Rabbit IgG (H+L) | Rabbit | Jackson ImmunoResearch Laboratories, Inc. (111-095-003) | 1:400 | |
| TRITC- AffiniPure Goat Anti-Mouse IgG (H+L) | Mouse | Jackson ImmunoResearch Laboratories, Inc. (115-025-003) | 1:400 | |
| Peroxidase-AffiniPure Goat Anti-Mouse IgG (H+L) | Mouse | Jackson ImmunoResearch Laboratories, Inc. (115-035-003) | | 1:5000 (for b-actin) 1:1000 (for cdc42) |
| Peroxidase-AffiniPure Goat Anti-Rat IgG (H+L) | Rat | Jackson ImmunoResearch Laboratories, Inc. (112-035-003) | | 1:1000 |
| Peroxidase-AffiniPure Goat Anti-Rabbit IgG (H+L) | Rabbit | Jackson ImmunoResearch Laboratories, Inc. (111-035-003) | | 1:5000 (for MST3) 1:1000 (for caspase3) |

permeablized with PBS/1% BSA/0.1% Triton X-100 for 30 min. The cells were incubated with primary antibodies for 1 h or overnight, followed by FITC- or TRITC-conjugated secondary antibodies for 1 h (Table 1). The cells were mounted to slides with gevetol medium and visualized using a Dragonfly High Speed Confocal Microscope System (Oxford Instruments) equipped with a FLUOTAR objective (25X, 0.95 NA, Oil) and an EMCCD camera (iXon Ultra 888) or Nikon camera. Images were acquired using Imaris software. Representative images were uniformly processed in Adobe Photoshop using the brightness and contrast tools.

## Cdc42 activity assay

A Cdc42 activation assay kit (Cytoskeleton Inc., Cat. #BK034) was used to perform the Cdc42 activation assay. Type I collagen diluted with $H_2O$ to 50 μg/ml was coated onto 100 mm culture dishes for 2 h. MDCK cells at 70% confluency were seeded on plastic plates or coated dishes for 24 h. The cells were washed in cold PBS and then lysed with lysis buffer. GTP-bound Cdc42 was affinity precipitated from cell lysates (1 mg of protein) using an immobilized GST fusion construct of the Cdc42 binding domain for 1 h. The complexes were then subjected to western blot analysis using Cdc42-specific antibodies. Total cellular lysates were also separated by SDS-PAGE, and western blot analysis with anti-Cdc42 antibodies was performed as a control for protein loading.

## Statistical analysis

Values are reported as the mean ± SD. Statistical analysis was performed in Microsoft Excel 2013 using one-way analysis of variance (ANOVA), followed by Bonferroni post hoc tests. *P values* < 0.05 were considered significant.

## Results

MST3 regulates multiple biological processes, including apoptosis [16], cell migration [17], and cytoskeletal rearrangements [22]. To assess MST3 activity in cyst formation, the lysine (K53) in the ATP-binding pocket of MST3 was mutated to arginine. Without ATP binding, the mutant MST3-K53R was a kinase dead (KD) MST3 with neither kinase activity nor autophosphorylation ability [17, 23]. Our previous research reported that HA-MST3 had higher activity than HA-MST3 through assaying radiolabeled [γ-$^{32}$P] ATP on MST3 peptide substrate [23]. To evaluate whether HA-tagged MST3 (HA-MST3) and HA-tagged MST3-K53R (HA-MST3-KD) were stably overexpressed in MDCK cells, HA antibody was used to detect overexpressing HA-MST3 and HA-MST3-KD in MDCK cells. Because MST3-KD could not be autophosphorylated, HA-MST3-KD migrated faster than HA-MST3 in SDS-PAGE (Fig 1, left panel). The MST3 antibody detected endogenous MST3 in control MDCK cells; however, the MST3 antibody not only detected endogenous MST3 but also HA-MST3 and HA-MST3-KD in both overexpressing HA-MST3 and HA-MST3-KD MDCK cells. Likewise, HA-MST3-KD migrated faster than HA-MST3 in SDS-PAGE (Fig 1, right panel). We further examined the activity of HA-MST3 and HA-MST3 KD through Phos-tag SDS/PAGE, a phosphate-affinity gel electrophoresis in which the phosphorylated proteins were delayed because they were trapped at phos-tag sites [24]. Again, HA antibody could not detect any bands in control cell (Fig 1B, upper and middle panels, lanes 1 and 2). HA-MST3 apparently shifted to a single band with a much slower mobility (Fig 1B, lane 3, arrow), which was less intensive presumably due to a serine/threonine phosphatase, PP2A, treatment (Fig 1B, lane 4, arrow). In contrast, HA-MST3 KD migrated at a position close to the unphosphorylated band, indicating that loss of kinase activity could not be phosphorylated (Fig 1B, upper panel, lanes 5 and 6). These results demonstrated that we successfully established overexpressed HA-MST3 and HA-MST3-KD in MDCK cells.

To assess epithelial morphogenesis, cells were grown in 3D collagen and Matrigel. At Day 11, control cells had formed cysts with well-polarized epithelial monolayers surrounding single hollow lumens in both collagen and Matrigel. HA-MST3 cells had smaller single hollow lumens with membrane blebs (Fig 2A, arrows). It was reported that as apoptosis progresses, blebs may break away from the cell body to form membrane-clad apoptotic bodies [25]. These blebs could be stained by caspase 3 in Fig 3A, indicating that HA-MST3 progressed significantly apoptosis at peripheral cells of cysts. The lumens were circled by dotted lines and the average area of cysts was approximately 41.3 μm$^2$ smaller than that of control cells (approximately 103 μm$^2$). In contrast, HA-MST3-KD cells displayed an irregular and multilumen phenotype (Fig 2A). Comparison the lumen formation at the 4$^{th}$ day of collagen culture, control cells displayed approximately 42% of single lumen and 36% of multilumen phenotype; that was progressively replaced approximately 75% of single lumen by the 11$^{th}$ day of culture (Fig 2B). In Matrigel, an extracellular matrix with strong external polarization cues [7] increased single lumen formation to approximately 63% and reduced multilumen formation to 25% at the 4$^{th}$ day of culture; by the 11$^{th}$ day of culture, approximately 80% of single lumen was formed with no significant increment. These results indicated that Matrigel accelerated single lumen formation and corrected multilumen phenotypes in collagen gels to single lumen phenotypes at Day 4 of culture, as reported in a previous study [7]. HA-MST3-overexpressing cells had slower cyst formation and fewer cysts. At the 4$^{th}$ day of collagen culture, HA-MST3 cells only displayed approximately 30% of single lumen and 22% of multilumen phenotypes, which were progressively replaced by approximately 44% of single lumen at 11 days of culture. Matrigel could also increase single lumen formation to approximately 46–50% from 4 to 11 days of culture. These results indicated that overexpressed MST3 inhibited cyst growth, consequently

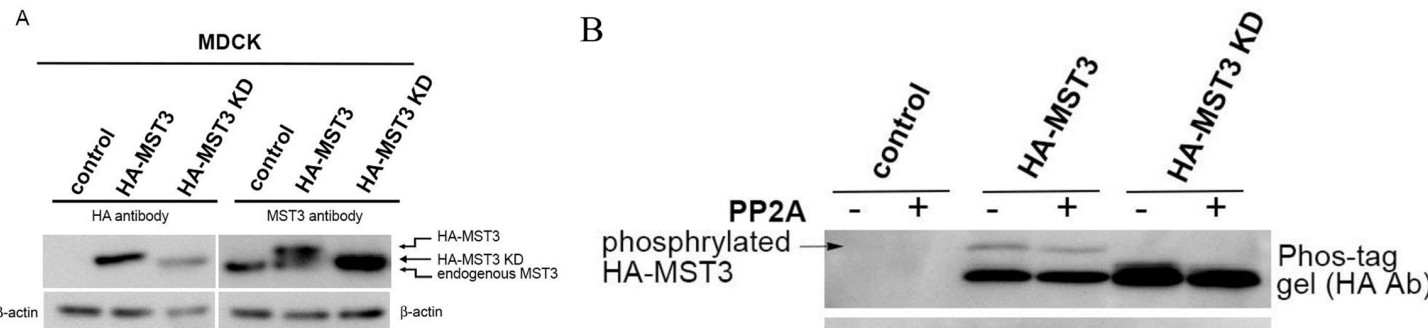

**Fig 1. Stable expression of HA-MST3 and HA-MST3 KD in MDCK cells.** MDCK (control), HA-MST3 and HA-MST3 KD transfectants were lysed (A) equal amounts of cell lysates were analyzed by immunoblot with anti-HA and anti-MST3 antibodies. (B) 50 μg cells lysates treated with or without PP2A were analyzed by Phos-tag gel or normal SDS/PAGE and immunoblot with anti-HA and actin antibodies. The bands shown as arrow were phosphorylated HA-MST3, which were trapped by phos-tag.

inhibiting cyst size. In contrast, at 4 days of collagen culture, HA-MST3-KD cells displayed only approximately 8% of single lumen and 90% of multilumen phenotype. The single lumen phenotype was not increased until 11 days of culture. Matrigel increased single lumen formation from 34% to 55% from Day 4 to Day 11 of culture. However, the percentage of multilumen cysts in both Matrigel and collagen was much higher than that in control and HA-MST3 cells (Fig 2B), indicating that loss of MST3 kinase activity caused multilumen formation.

Apoptosis is a control mechanism that ensures the clearance of cells in the lumen when the cell density is high in collagen [7] (Fig 3A). Since Matrigel has a strong growth signal to form cysts, we used collagen to examine whether MST3-mediated apoptosis was involved in cyst formation. The control cells displayed apoptosis in the central cells of the lumen (circled by dotted lines) at Day 4 of culture by cleaved caspase3 staining. At Day 8 of culture, fewer apoptotic cells were detected by caspase3 staining, indicating that central cells have already processed apoptosis to form the lumen. HA-MST3 cells displayed apoptosis in the central cells of the lumen at Day 4 of culture; however, apoptotic particles were intensively stained at peripheral cells of cysts at Day 8 of culture (Fig 3B). These apoptotic particles, which were also observed under a light microscope, presented as blebs around the cyst (Fig 2A, arrows). The HA-MST3-KD cells still underwent apoptosis in the central cells of the lumens, indicating that the multilumen phenotype of HA-MST3 KD may not be caused by apoptosis resistance. To further examine whether HA-MST3 exhibited higher apoptotic activities, we used western blotting to examine cleaved caspase3 expression. Fig 3C shows that HA-MST3 cells exhibited the highest caspase activities, and HA-MST3-KD cells exhibited the lowest caspase activities both with and without collagen coating. On the other hand, the regulation of proliferation is also crucial to lumen formation. Ki67, a proliferation marker, was observed in peripheral cells instead of in the central cells of the cysts in control cells. HA-MST3-overexpressing cells displayed less Ki67 in peripheral cells than control cells. However, Ki67 was observed in the central cells of the lumen (Fig 3D, arrows) in HA-MST3 KD cells, indicating that persistent proliferation might lead to a multilumen phenotype of HA-MST3-KD cells (Fig 3D). Taken together, the cells overexpressing MST3 had a slower growth rate with higher caspase 3 activity, subsequently forming smaller cysts. In contrast, loss of MST3 activity with persistent proliferation caused multilumen formation. The MST3 activity contributes to cyst formation by balancing apoptosis and proliferation.

Next, we examined the cell polarity of these cysts by staining apical ZO-1 and basolateral E-cad. At Day 4 and Day 8 of Matrigel culture, ZO-1 (green) was present at the apical site, and E-cad (red) was present at the basolateral site of the cyst in both control (Fig 4A, c and f) and

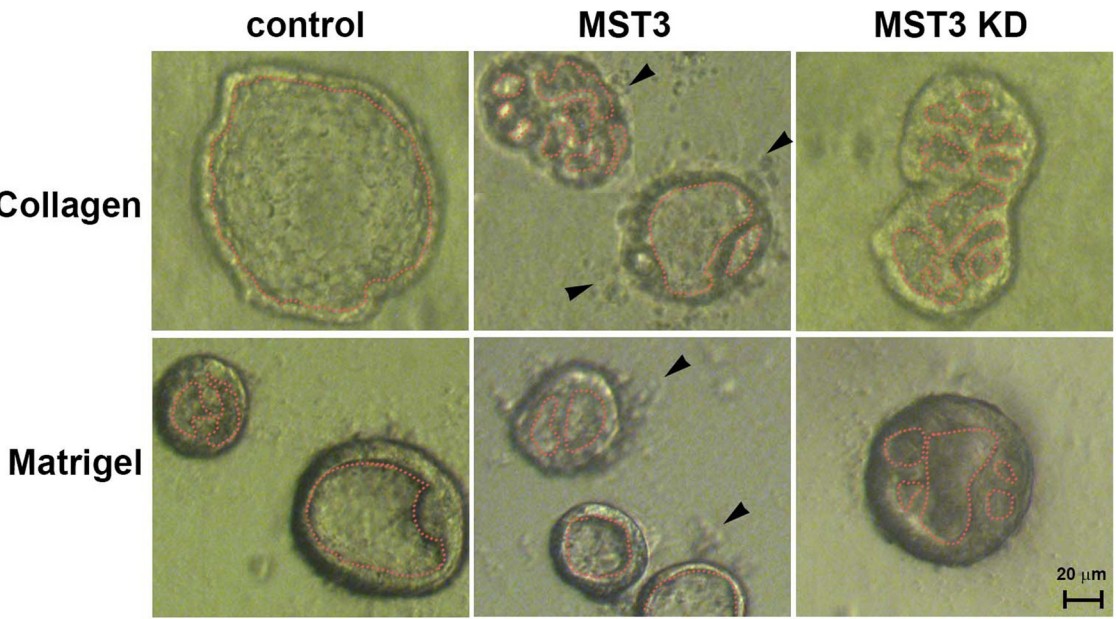

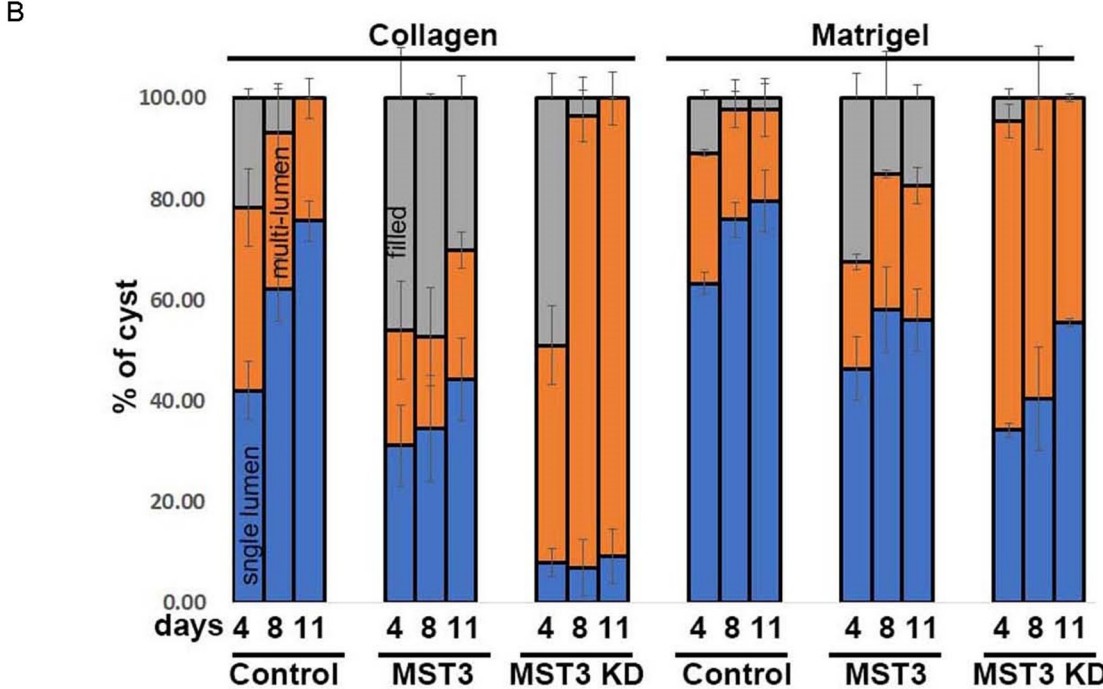

**Fig 2. Effect of MST3 on cyst progression in the MDCK cyst model.** (A) Representative images of control, HA-MST3 and HA-MST3 KD cysts were taken at Days 4, 8 and 11 after seeding in collagen or Matrigel. The dotted line circled the lumens; HA-MST3 KD developed more mutilumen cysts than control and HA-MST3 cells, which were replaced more single lumen in Matrigel. The arrows in HA-MST3 cells denoted apoptotic bodies (B) Lumen phenotypes at each time point in both growth conditions were classified as filled (no lumen; gray), multilumen (orange), or single lumen (blue). The mean ± SD of >30 cysts from each experiment is shown, for three independent experiments.

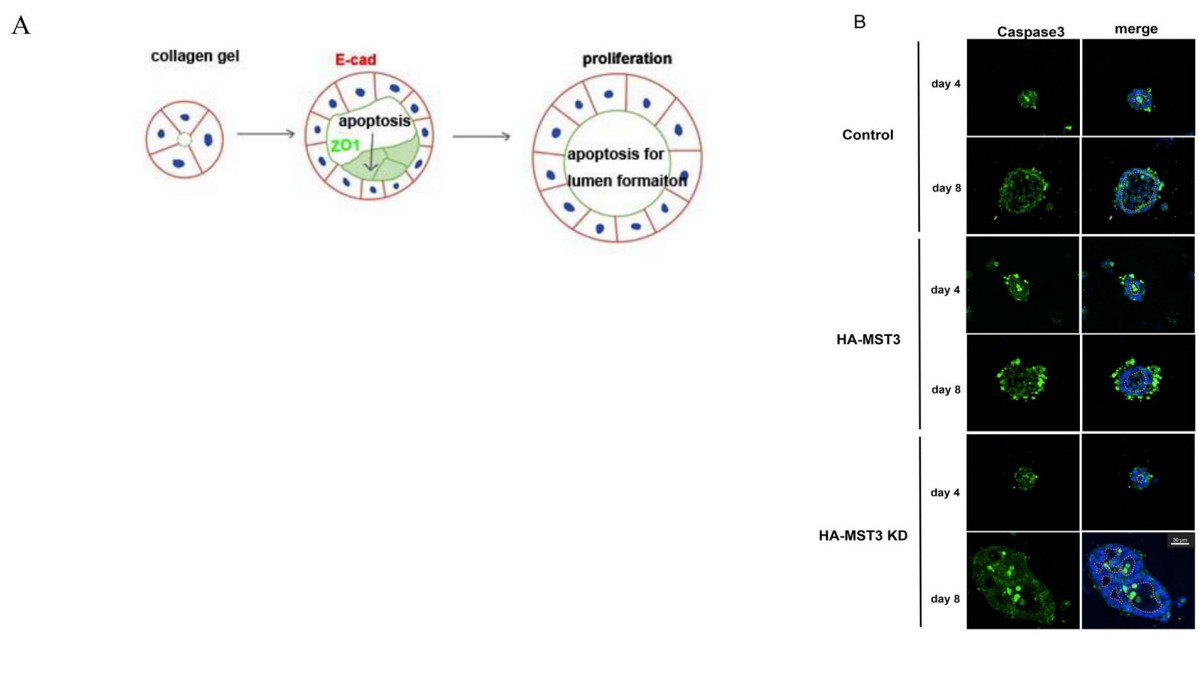

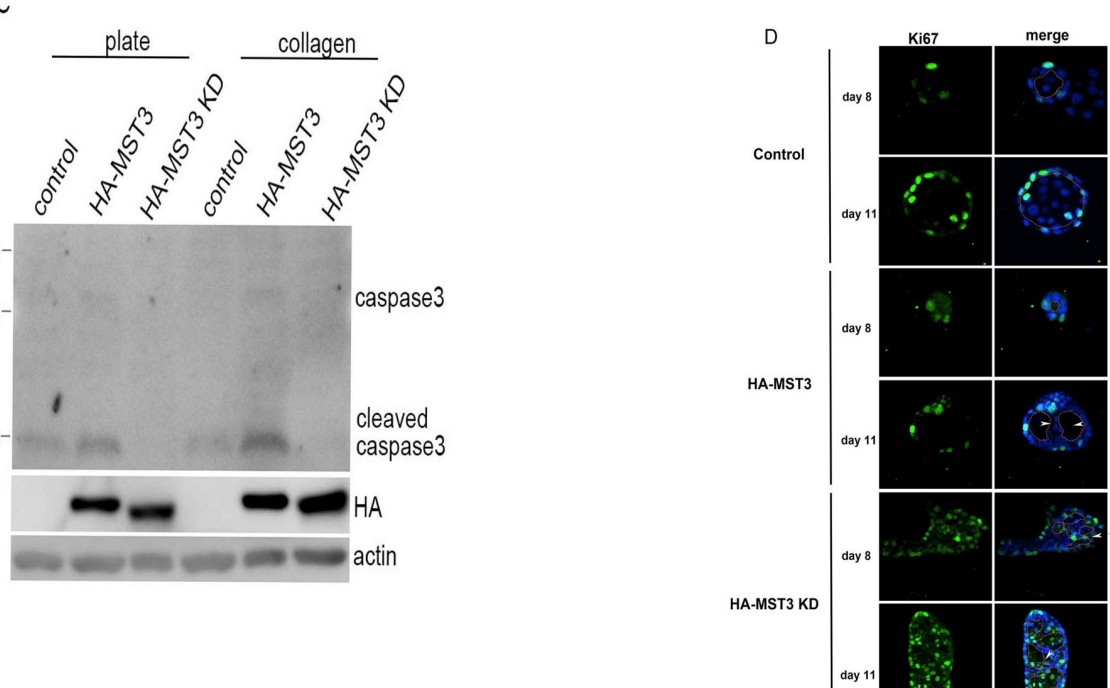

**Fig 3. The apoptosis and proliferation effect of MST3 on cyst progression.** (A) To form cysts, epithelial cell is polarized into an apical and a basolateral plasma membrane domain. Apoptosis is needed that ensures the clearance of cells in the lumen when the cells are in collagen. Proliferation is needed for cyst growth (7). Representative images of control, HA-MST3 and HA-MST3 KD cysts grown in collagen for 4 and 8 days. The dotted line circled the lumens. (B) Cells were fixed and stained to detect nuclei (blue) and cleaved caspase3 (green). The control cells displayed apoptosis in the central cells of the lumen (circled by dotted lines) at Day 4 of culture by cleaved caspase3 staining. HA-MST3 cells displayed apoptosis in the central cells of the lumen at Day 4 of culture; however, apoptotic particles were intensively stained at peripheral cells of cysts at Day 8 of culture. The HA-MST3-KD cells still underwent apoptosis in the central cells of the lumens. (C) Cells were lysed, and equal amounts of cell lysates were analyzed by immunoblot with a cleaved caspase-3 antibody. (D) Cells were fixed and stained to detect nuclei (blue) and Ki67 (green). Ki67 was observed in the central cells of the lumen in HA-MST3 KD cells Bar, 30 μm.

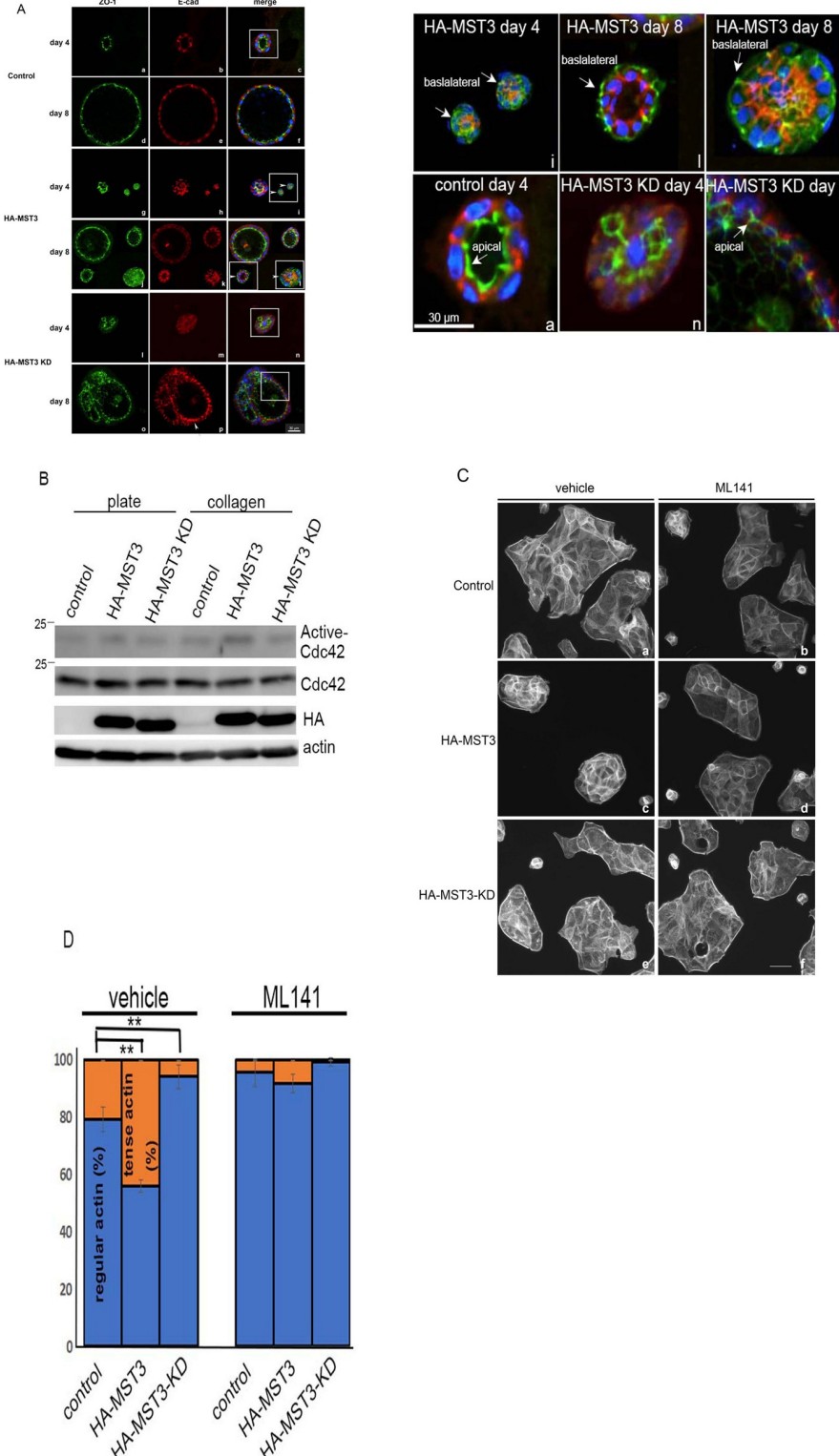

**Fig 4. Effect of MST3 on Cdc42 activity.** (A) Representative images of control, HA-MST3 and HA-MST3 KD cysts grown in Matrigel for 4 or 8 days. Cells were fixed and stained to detect nuclei (blue), ZO-1 (green) and E-cadherin (red). The white solid-line boxed areas were enlarged and shown below. The arrows indicated ZO-1 localization (green). ZO1 was present at apical site compared with the E-cadherin at basolateral site (red) in control cell and HA-MST3 KD cells; however, ZO1 in HA-MST3 was still localized at basolateral site. (B) Control, HA-MST3, and

HA-MST3 KD cells were lysed, and 1 mg cell lysate was incubated with Cdc42 binding proteins. The total Cdc42 from cell lysate (Cdc42) and Cdc42 binding protein bound Cdc42 (active-Cdc42) was analyzed by immunoblot with Cdc42 antibody. (C) Representative images of $5 \times 10^4$ cells on coverslips treated with or without 10 μM ML141 for 2 h and stained with FITC-phalloidin. Bar, 20 μm. (D) Quantification of belt-like actin >500 cells per group. For all measurements, the group averages are shown, and the error bars represent SD.

HA-MST3-KD cells (Fig 4A, n and q). Surprisingly, ZO-1 was present at the basal site in a few HA-MST3-overexpressing cysts on Days 4 and 8 of culture (Fig 4A, i and l, arrows). The enlarged images of white boxed area were shown below. These results indicated that overexpression of HA-MST3 retarded ZO-1 translocation to the apical site. Since Cdc42 regulates ZO-1 through Tuba at the apical margin of the epithelial cell junctions [26] and reduced Cdc42 by siRNA had multiple lumens with decreased F-actin [10], we examined Cdc42 activity by PAK-PBD pull down. We observed that more activated Cdc42 was pulled down in HA-MST3 cells than in control and HA-MST3 KD cells on collagen-coated plates (Fig 4B). Since activation of Cdc42 has been shown to increase actin accumulation at cell junctions [27], we also examined Cdc42 activity by observation of FITC-phalloidin staining. The cells with MST3 overexpression were smaller and exhibited approximately 45% tense actin, which was significantly higher than the control cells with approximately 20% tense actin (p<0.004). The tense belt-like actin of HA-MST3 cells and control cells was reduced to approximately 8% and 4%, respectively, with ML141, a Cdc42 inhibitor treatment. In contrast, the belt-like actin was loose and irregular in MST3-KD-overexpressing cells compared with control cells (Fig 4C, e). These results indicated that overexpression of MST3 resulted in higher Cdc42 activity with a tense actin belt.

The F-actin tension at the cell surface is regulated by Cdc42-mediated ZO-1, which affects lower E-cadherin distribution [26]. To investigate whether MST3-mediated ZO-1 was involved in cyst formation, we examined cell–cell junctions by a $Ca^{2+}$ switch model. When the cells were incubated under low $Ca^{2+}$ concentrations, the cells exhibited incomplete development of apical-basolateral polarity (data not shown). After the addition of calcium for 1 h, E-cad and ZO-1 were concentrated at primordial spot-like junctions in all control, HA-MST3 and HA-MST3-KD cells; however, increased amounts of ZO-1 and E-cad were observed in the regions where multiple cells formed contacts in HA-MST3 cells (Figs 5A, d, arrows and S1). After the addition of calcium for 4 h, the polarity of control cells was re-established, accompanied by a relocalization of ZO-1 and E-cad at tight junctions and adherens junctions of the plasma membrane, respectively. However, the relocalization of ZO-1 and E-cad occurred rapidly in HA-MST3-KD cells. Closer observation of those cells with HA-MST3-KD overexpressing revealed that in some cases E-cad showed slightly more lateral spread (Fig 5B, h, arrows). More lateral spread of E-cad was also observed in HAMST3-KD cysts (Fig 4A, p, arrows). In contrast, increased amounts of ZO-1 were still immobilized, followed by E-cad distribution; consequently, the relocation of ZO-1 and E-cad were delayed in HA-MST3 cells (Figs 5B, d, arrows and S1). These results indicated that MST3 inhibited ZO-1 and E-cad trafficking to apical and basolateral sites, respectively, which may consequently delay cyst formation. Loss of MST3 activity may accelerate ZO-1 trafficking, and, consequently, multilumen cysts may form.

Since we found that MST3 primarily localized at the basal site of collecting ducts in SHRs (spontaneous hypertensive rats), MST3 was present in the cytosol of collecting ducts in WKYs (as control rats) [19]. We examined whether MST3 activity affected its localization by using HA antibody staining. We found that HA-MST3 localized primarily in the cytosol of the cysts at Day 4 of culture. Take a closer look at the white boxed area, HA tagged MST3 (green) were away from apical ZO1 (red) and were localized at the cytosol. Although some MST3

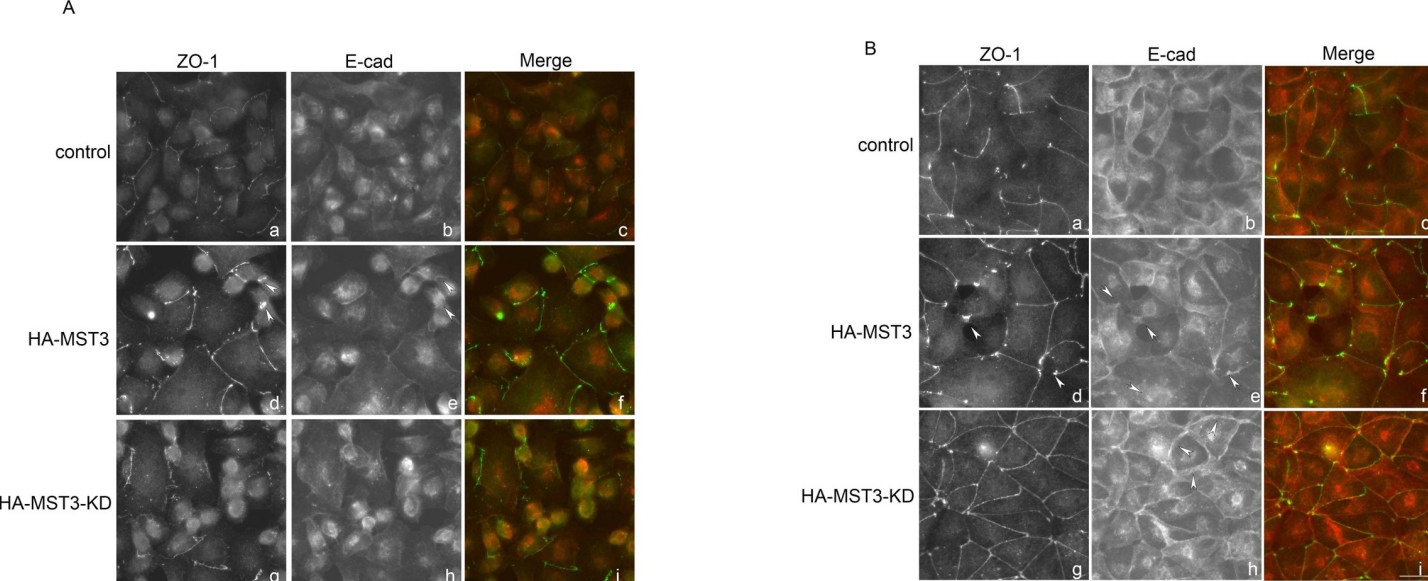

**Fig 5. Effect of MST3 on ZO-1 and E-cad localization on tight junction formation.** Representative images of $3 \times 10^5$ control, HA-MST3, and HA-MST3 KD cells grown on 2 mg/ml collagen-coated coverslips. After 72 h, the medium was changed to medium that was depleted of calcium for 24 h. Cells were fixed at (A) 1 h and (B) 4 h after the readdition of calcium and stained for ZO-1 (green) and E-cad (red). Bar, 20 μm.

translocated to the subapical sites close to ZO-1 in the cysts, some MST3 was still present in the cytosol and basal site of the cysts at Day 8 of culture (Fig 6, c and f). The majority of kinase-dead MST3 were localized at subapical close to apical ZO1, at Days 4 and 8 of culture (Fig 6, i and l). These results indicate that MST3 might have to be deactivated during cyst formation and then MST3 could be delivered to subapical sites.

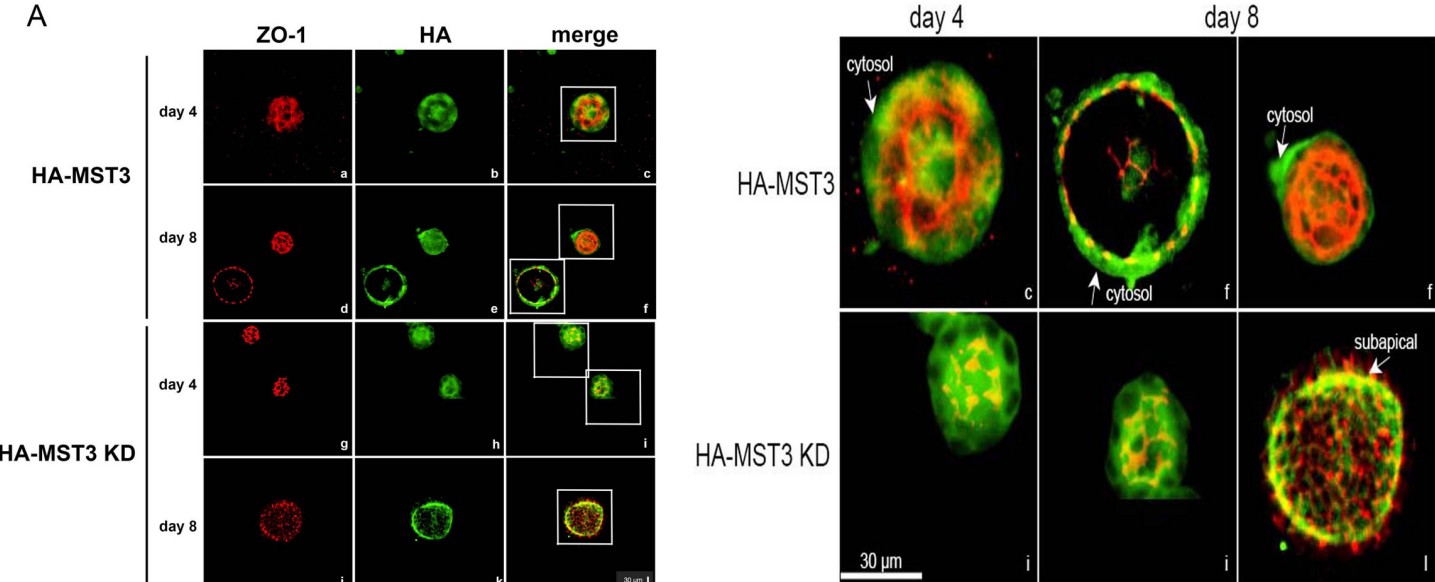

**Fig 6. MST3 localization during cyst progression.** (A) Representative images of HA-MST3 and HA-MST3 KD cysts grown in Matrigel for 4 or 8 days. Cells were fixed and stained to detect ZO-1 (red) and HA-MST3 (green). The white solid-line boxed areas were enlarged and shown below. The arrows indicated HA tagged proteins (green) localization. The HA-MST3 were away from ZO1 (red), localized at the cytosol; HA-MST3 KD were close to ZO1, localized at subapical.

## Discussion

Coordination of fluid accumulation, membrane delivery, cell proliferation and apoptosis is required for epithelial polarization and cyst and tubule formation. Among the diverse roles of MST3, we have defined its roles in the regulation of renal ion channels to maintain stable blood pressure [18, 19, 28], cell apoptosis [16] and cell migration [17]. These functions of MST3 are all involved in cell polarity and cyst formation. The findings presented here identify a new link between Cdc42 GTPase and MST3 that functions in the establishment of cell polarity in MDCK cells. We showed that cells with kinase-dead MST3 had lower Cdc42 activity and exhibited multilumen cysts; however, cells overexpressing MST3 had higher Cdc42 activity and exhibited smaller and fewer cysts than WT MDCK cells. The overactivated MST3 localizes at cytosol and kinase-dead MST3 localized at subapical site of cysts. These results indicate that MST3 might have to be deactivated to regulate Cdc42 activity during cyst formation. Then MST3 and Cdc42 could be delivered to subapical sites.

Cdc42 accumulates at cell-cell contact sites to regulate cell-cell adhesion, which is the first step of cell polarization [29]. To drive lumen formation in 3D culture of MDCK cells, Cdc42 regulates the exocytosis vesicles containing apical markers to the apical membrane, and then Cdc42 is recruited to the apical membrane. However, the constitutive activation of Cdc42 induces ZO-1 localization close to basal sites instead of at apical sites [9, 30, 31]. Cdc42 activity is induced by the production of $PIP_2$ (PtdIns(4,5)$P_2$. Cdc42 and $PIP_2$ can stimulate actin polymerization to regulate actin polymerization and early mammalian development of [32]. $PIP_2$ is apically enriched, whereas $PIP_3$ is basolateral. The asymmetry PIP (phosphatidylinositol phosphate) plays a role in apical-basal polarization in MDCK cysts [33]. When exogenous PIP2 is added to the basal membrane, Cdc42 and ZO-1, along with other tight junction proteins, relocalize to the basal sites of the cyst [10]. Taken together, these results indicate that ZO-1 localization is regulated by Cdc42 localization. Our results are consistent with previous studies and show that ZO-1 localized at basal sites of cysts in MST3-overexpressing MDCK cells (Fig 4A) with higher Cdc42 activity (Fig 5). Although ZO-1 was at basal sites of the cysts, the lumen was still formed, indicating that overexpressed MST3 delayed ZO-1 targeting to the apical sites (Fig 4A).

The retarded localization of ZO-1 to the apical site of cysts in HA-MST3 cells was also observed in the $Ca^{2+}$ switch assay. We observed that ZO-1 and E-cad were still immobilized at high tension cell edges or vertices after $Ca^{2+}$ repletion for 4 h in HA-MST3 cells (Fig 5B, d and e, arrows). In contrast, E-cad was rapidly sorted to the adherens junctions in HA-MST3-KD cells. Although E-cad was rapidly sorted to adherens junctions, they were loose to spread to the cytosol. The loose F-actin tension (Fig 4C, e) was consistent with the results of loose ZO-1 and E-cad (Fig 5B, g and h). Our previous study showed that downregulation of MST3 by siRNA decreased E-cad expression at the migrating cell edge of MCF7 cells, leading to cell migration. This previous study showed that MST3 was involved in E-cad regulation by protein-tyrosine phosphatase (PTP)-PEST [17]. Taken together, MST3 regulates cell-cell junctions, affecting cell migration and cell polarity.

In the kidney, our study showed that MST3 localized at the subapical site and cytosol of the renal tubules [18, 19]. Although the MST3 knockdown mice exhibited normal renal tubule morphology, more ENaC was present at apical sites than in WT mice, indicating that MST3 inhibited ENaC expression at apical sites in the kidney [18]. Here, we found that overexpressed MST3 localized close to the basal membrane instead of the subapical membrane of cysts. In contrast, loss of MST3 activity localized close to the subapical membrane of the cysts, in which ZO-1 also localized at the apical site of the cysts (Fig 6). ENaC activities could be increased by RhoA-mediated increases in the movement of ENaC to the membrane, which depended on microtubules [34]. Our results indicated that overexpressed HA-MST3 had higher Cdc42

activity and that kinase-dead MST3 had lower Cdc42 activity. These results might be associated with ENaC transport in renal tubules. Whether MST3-mediated Cdc42 or other GTPases are involved in ENaC and target protein destination needs further investigation.

YAP and its paralog TAZ are proliferation-activating transcriptional coactivators that shuttle between the nucleus and cytoplasm. The MST1- and MST2-mediated Hippo pathway regulates YAP and TAZ phosphorylation, resulting in suppression of YAP and TAZ nuclear localization, which is involved in organ size and cell growth [35]. YAP/TAZ nuclear localization is suppressed by tense circumferential actin belt tension, which inhibits cell proliferation [36]. We found that the cysts of HA-MST3 cells were smaller than those of control cells (Fig 2). HA-MST3-overexpressing cells exhibited stronger stress fibers than HA-MST3-KD cells (Fig 5A). In future studies, it will be important to define whether MST3 may also be involved in the Hippo pathway to regulate organ size through cell junction regulation.

## Supporting information

**S1 Fig. Effect of MST3 on ZO-1 and E-cad localization on tight junction formation in the Ca$^{2+}$ switch assay, related to Fig 5.** Representative images of $3 \times 10^5$ control, HA-MST3, and HA-MST3 KD cells grown on collagen-coated coverslips with large field. After 72 h, the medium was changed to medium that was depleted of calcium for 24 h. Cells were fixed at (A) 1 h and (B) 4 h after the readdition of calcium and stained for ZO-1 (green) and E-cad (red) with X20.
(JPG)

## Acknowledgments

The confocal laser microscopy was performed and provided by the Medical Research Core Facilities Center, Office of Research & Development at China medical University, Taichung, Taiwan.

## Author Contributions

**Conceptualization:** Pei Lin, Te-Ling Lu.

**Data curation:** Chee-Hong Chan, Tse-Yen Yang, Jhen-Yang Jhong, Hui-Fen Cheng.

**Investigation:** Bo-Ying Bao, Jhen-Yang Jhong, Te-Hsiu Lee.

**Methodology:** Tse-Yen Yang, Yui-Ping Weng.

**Resources:** Bo-Ying Bao.

**Supervision:** Te-Ling Lu.

**Writing – original draft:** Te-Ling Lu.

**Writing – review & editing:** Chee-Hong Chan, Te-Ling Lu.

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
