## [Decision Letter · Decision Letter 0]

23 Mar 2023

PONE-D-23-00666Epithelial polarization in the 3D matrix requires MST3 signaling to regulate ZO-1 positionPLOS ONE

Dear Dr. Lu,

Thank you for submitting your manuscript to PLOS ONE. After careful consideration, we feel that it has merit but does not fully meet PLOS ONE’s publication criteria as it currently stands. Therefore, we invite you to submit a revised version of the manuscript that addresses the points raised during the review process.

Thank you for your patience while your manuscript was peer-reviewed at PLoS One.  Please accept my apologies for the delay in providing you with our decision.  The manuscript has now been evaluated by two independent reviewers.

The reviews are attached below.  You will see that both of the reviewers find the study is well-designed and conducted thoroughly but they think there are several concerns.  Please make sure to address all the points raised by the reviewers in the revised manuscript.

We look forward to receiving your revised manuscript.

Kind regards,

Tomohito Higashi, Ph.D.

Academic Editor

PLOS ONE

Journal Requirements:

"This work was supported by grants from the National Science Council of Taiwan research MOST 109-2320-B-039-017, China Medical University Grant CMU110-MF-76, Chang Bing Show-Chwan Memorial Hospital Grant BRD109001, and An Nan Hospital-China Medical University in Taiwan Grant ANHRF111-10."

Reviewers' comments:

Reviewer's Responses to Questions

**Comments to the Author**

1. Is the manuscript technically sound, and do the data support the conclusions?

Reviewer #1: Yes

Reviewer #2: Partly

2. Has the statistical analysis been performed appropriately and rigorously? 

Reviewer #1: Yes

Reviewer #2: No

3. Have the authors made all data underlying the findings in their manuscript fully available?

Reviewer #1: Yes

Reviewer #2: Yes

4. Is the manuscript presented in an intelligible fashion and written in standard English?

Reviewer #1: Yes

Reviewer #2: Yes

5. Review Comments to the Author

Reviewer #1: Chan and colleagues present a study investigating the role of MST3 kinase in regulation of cell adhesion, junctional architecture, and cyst formation. Through investigation of cytoskeletal architecture, the authors identify Cdc42 as an effector of MST3 that links MST3 activity to cytoskeletal organization. Overall, this study is well-focused using appropriate assays, but some concerns remain prior to suggesting approval for publication:

The authors use distribution in SDS-PAGE as a surrogate for demonstrating lack of autophosphorylation in the KD construct, but it is possible that an issue in cloning or protein processing led to the observed distribution in SDS-PAGE rather than the lack of kinase activity. Is there a way to more directly demonstrate that the KD construct indeed lacks the ability to phosphorylate?

How are lumens identified in Fig. 2? In the representative images, it is difficult to determine where lumens are forming in the collagen culture condition, and it is not immediately apparent that there is a difference between the lumen formation in the MST3 and MST3-KD conditions. Also, the black arrows are not described in the figure caption.

Figure 6 seems overinterpreted based on the images provided. It is also not clear what the significance of the authors’ interpretation is with regard to distribution. In the images provided at the magnification presented, it is difficult to determine the relative distribution of ZO-1 and the HA tag.

The introduction and discussion are written from a cell biology perspective, which is understandable given the focus of the study, however some perspective on potential translation would be justified. Are there any implications for MST3 or the other MST kinases in disease pathogenesis?

Reviewer #2: In this manuscript, Chan et al generated MDCK cell lines stably expressing HA tagge MST3 or kinase-dead MST3 (MST3-KD) with a point mutation K53R. The they assessed cyst formation, levels of caspase 3 (for appotptosis) and Ki67 (for proliferation), location of junction protein ZO-1 and Ecadherin, CDC42 activation, and Phalloidin (for actin bundle). The experimental design is straight forward, but the data (figure) presentation is not very reader friendly, separate different panels in different page with legend under the last panel page make it a bit difficult to catch the main message of the whole figure.

Overall the data does go along with the conclusion but is not very strong in support the conclusion.

1. Fig. 2A panel are selected images, but do not reflex the full picture of cyst distribution on dish, quantification of the size distribution of cyst son gel could help provide more information. For cell culture studies, 3 repeats is minimal requirement, but not rigorous enough.

2. Fig. 3 ideally higher resolution 3-d reconstruct of cyst staining are need to quantify the percentage of cleaved caspase 3 positive cells. Similar data showed be provided for Ki67 staining. The Ki67 in MST3-KD is probably most striking data.

3. Fig 4A, higher resolution images for Cell-cell interaction level are need to go with these images at whole cyst level. ML-141 should be clearly introduced in main text.

4. Fig. 5 MDCK as polarized cell, studied junction protein distribution in cell culture on cove slips not necessary reflect physiological conditions. These data should generated from 3-D culture cells or cell culture on membranes/matrix which support polarization of epithelial cells.

5. Fig 6, it is not clear at current resolution, where MST3 locates in terms subcellular level. Ideally, to more accurately determine MST3 location, the expression of MST3 level should be titrated at a lower level to better reflect endogenous conditions.

6. PLOS authors have the option to publish the peer review history of their article (what does this mean?). If published, this will include your full peer review and any attached files.

Reviewer #1: No

Reviewer #2: No

---

## [Author Response · Author response to Decision Letter 0]

6 Apr 2023

Reviewer 1:

Thank you for the comments.

1. Is there a way to more directly demonstrate that the KD construct indeed lacks the ability to phosphorylate?

Activated MST3 stimulated MST3 autophosphorylation in several serine/threonine sites (DOI: 10.1042/BJ20112000). We previously reported that MST3 activity could be measured by the detection of the ability of MST3 to transfer [γ-32P]-ATP to an MST3 peptide substrate. We found that HA-MST3 had higher activity than HA-MST3 K53R (HA-MST3 KD) (Journal of inorganic biochemistry 160 (2016) 33-39). Here, we further used phos-tag gel to detect phosphorylated HA-MST3 and KD phosphorylation to determine their activity. We added the figure 1B and explained the results in the main text as: 

We further examined the activity of HA-MST3 and HA-MST3 KD through Phos-tag SDS/PAGE, a phosphate-affinity gel electrophoresis in which the phosphorylated proteins were delayed because they were trapped at phos-tag sites (25). Again, HA antibody could not detect any bands in control cell (Fig. 1B, upper and middle panels, lane 1 and 2). HA-MST3 apparently shifted to a single band with a much slower mobility (Fig. 1B, lane 3, arrow), which was less intensive presumably due to a serine/threonine phosphatase, PP2A, treatment (Fig. 1B, lane 4, arrow). In contrast, HA-MST3 KD migrated at a position close to the unphosphorylated band, indicating that loss of kinase activity could not be phosphorylated (Fig. 1B, upper panel, lane 5 and 6). 

2. How are lumens identified in Fig. 2? In the representative images, it is difficult to determine where lumens are forming in the collagen culture condition, and it is not immediately apparent that there is a difference between the lumen formation in the MST3 and MST3-KD conditions. Also, the black arrows are not described in the figure caption.

Thank you for the comments.

We circled the lumen with dotted lines in fig. 2A. HA-MST3 cells did develop approximately 25% multilumen cysts in collagen gel until 11 days; HA-MST3-KD cells displayed only approximately 8% of single lumen and 90% of multilumen phenotype. The single lumen phenotype was not increased until 11 days of culture. However, the percentage of multilumen cysts of HA-MST3 KD in both Matrigel and collagen was much higher than that in control and HA-MST3 cells (Fig. 2B). Because multilumen cysts were formed in both HA-MST3 and HA-MST3 KD cells, it is not immediately difference in representative fig. 2A. We calculated the percentage of single, multilumen and filled cysts in both Matrigel and collagen in fig. 2B. The multilumen phenotype of HA-MST3 KD was much higher than that in control and HA-MST3 cells (Fig. 2B), indicating that loss of MST3 kinase activity caused multilumen formation.

Also, the black arrows are not described in the figure caption.

We added “The arrows denoted apoptotic bodies” in figure legend”. We also added “As apoptosis progresses, blebs may break away from the cell body to form membrane-clad apoptotic bodies (doi:10.1038/cdd.2013.69). These blebs could be stained by caspase 3 in Fig 3A” in results.

3. Figure 6 seems overinterpreted based on the images provided. It is also not clear what the significance of the authors’ interpretation is with regard to distribution. In the images provided at the magnification presented, it is difficult to determine the relative distribution of ZO-1 and the HA tag.

The reason to show the localization of MST3 due to our previous research. In mice and Rat, we found that MST3 localization was important for channel activity (DOI: 10.1007/s11255-018-2011-x, doi.org/10.1152/ajprenal.00455.2018). 

We revised figure 6 with higher resolution and labeled the localization of HA-tagged MST3. We explained the results in the main text and figure legend shown below. 

Main text:

Take a closer look at the white boxed area, HA tagged MST3 (green) were away from apical ZO1 (red) and were localized at the cytosol. Although some MST3 translocated to the subapical sites close to ZO-1 in the cysts, some MST3 was still present in the cytosol and basal site of the cysts at Day 8 of culture (Fig. 6, c and f). The majority of kinase-dead MST3 were localized at subapical close to apical ZO1, at Days 4 and 8 of culture (Fig. 6, i and l). These results indicate that MST3 might have to be deactivated during cyst formation and then MST3 could be delivered to subapical sites.

Figure legend

The white solid-line boxed areas were enlarged and shown below. The arrows indicated HA tagged proteins (green) localization. The HA-MST3 were away from ZO1 (red), localized at the cytosol; HA-MST3 KD were close to ZO1, localized at subapical. 

4. The introduction and discussion are written from a cell biology perspective, which is understandable given the focus of the study, however some perspective on potential translation would be justified. Are there any implications for MST3 or the other MST kinases in disease pathogenesis?

Thank you for your comments again. 

Our previous research showed that MST3 play a role in renal hypertension through ENaC and NKCC2 regulation. Because these channels are localized at renal apical sites, we hypothesize that MST3 may regulate channels translocation. We introduced MST3 function in the last paragraph of introduction part. In addition, MST1 is important in Hippo-YAP pathway to control organ size. We suspected that MST3-involved cell polarity might also play a role in Hippo pathway. We discussed this in the last paragraph in the discussion part.

 

Reviewer #2: 

Thank you for the comments.

1. Fig. 2A panel are selected images, but do not reflex the full picture of cyst distribution on dish, quantification of the size distribution of cyst son gel could help provide more information. For cell culture studies, 3 repeats is minimal requirement, but not rigorous enough.

Thank you for the comments.

We circled the lumen with dotted lines in fig. 2A. In fig. 2B, we counted the cysts from at least 3 cell cultures and measured the percentage of lumen phenotype. We also measured the size of the cysts on day 11 and described in the results as “the average area of cysts was approximately 41.3 μm2 smaller than that of control cells (approximately 103 μm2). In contrast, HA-MST3-KD cells displayed an irregular and multilumen phenotype”. The HA-MST3-KD developed irregular multilumen cysts, therefore, the variance was significantly high. 

We calculated the percentage of single, multilumen and filled cysts in both Matrigel and collagen. The multilumen cysts of HA-MST3 KD was much higher than that in control and HA-MST3 cells (Fig. 2B), indicating that loss of MST3 kinase activity caused multilumen formation. The figure legend showed that “The mean ± SD of >30 cysts from each experiment is shown, for three independent experiments”.

2. Fig. 3 ideally higher resolution 3-d reconstruct of cyst staining are need to quantify the percentage of cleaved caspase 3 positive cells. Similar data showed be provided for Ki67 staining. The Ki67 in MST3-KD is probably most striking data.

The polarity, apoptosis and cell proliferation are essential during cyst formation, therefore, we added Fig. 3A and figure legend as “To form cysts, epithelial cell is polarized into an apical and a basolateral plasma membrane domain. Apoptosis is needed that ensures the clearance of cells in the lumen when the cells are in collagen. Proliferation is needed for cyst growth (7)”. We also added dotted line to show the apoptosis in central lumen in control and HA-MST3 KD cells. In contrast, apoptosis was present at the peripheral cells in HA-MST3 cells.

The legend of fig3, we revised as 

“The apoptosis and proliferation effect of MST3 on cyst progression. (A) To form cysts, epithelial cell is polarized into an apical and a basolateral plasma membrane domain. Apoptosis is needed that ensures the clearance of cells in the lumen when the cells are in collagen. Proliferation is needed for cyst growth (7). Representative images of control, HA-MST3 and HA-MST3 KD cysts grown in collagen for 4 and 8 days. The dotted line circled the lumens. (B) Cells were fixed and stained to detect nuclei (blue) and cleaved caspase3 (green). The control cells displayed apoptosis in the central cells of the lumen (circled by dotted lines) at Day 4 of culture by cleaved caspase3 staining. HA-MST3 cells displayed apoptosis in the central cells of the lumen at Day 4 of culture; however, apoptotic particles were intensively stained at peripheral cells of cysts at Day 8 of culture. The HA-MST3-KD cells still underwent apoptosis in the central cells of the lumens. (C) Cells were lysed, and equal amounts of cell lysates were analyzed by immunoblot with a cleaved caspase-3 antibody. (D) Cells were fixed and stained to detect nuclei (blue) and Ki67 (green). Ki67 was observed in the central cells of the lumen in HA-MST3 KD cells Bar, 30 μm.”

The results were described in the 3rd paragraph of result part.

3. Fig 4A, higher resolution images for Cell-cell interaction level are need to go with these images at whole cyst level. ML-141 should be clearly introduced in main text.

We revised figure 4 with higher resolution and we added “ML141, a Cdc42 inhibitor treatment” in the main text..

4. Fig. 5 MDCK as polarized cell, studied junction protein distribution in cell culture on cove slips not necessary reflect physiological conditions. These data should generated from 3-D culture cells or cell culture on membranes/matrix which support polarization of epithelial cells.

As reviewer’s comments, coverslip could not reflect junction and polarization. The polarization study should be performed on matrix-coating membrane or coverslip. Nigam. et al indicated that intracellular Ca2+ during the switch was related to the development of junctions and polarity of MDCK cells on collagen-coated coverslip (PNAS 1992 Vol. 89, pp. 6162-6166). Pia et al also used collagen to investigate the localization of MT1-MMP at the basal surface upon HGF treatment (DOI: 10.26508/lsa.201800276). We also examined the cell polarization on 2 mg/ml collagen-coated coverslip and described it in the figure legend and methods.

5. Fig 6, it is not clear at current resolution, where MST3 locates in terms subcellular level. Ideally, to more accurately determine MST3 location, the expression of MST3 level should be titrated at a lower level to better reflect endogenous conditions..

We revised figure 6 with higher resolution and labeled the localization of HA-tagged MST3. We explained the results in the main text and figure legend shown below.

Main text:

The white solid-line boxed areas were enlarged. The arrows indicated that HA tagged MST3 (green) were away from apical ZO1 (red), which were localized at the cytosl. Although some MST3 translocated to the subapical sites close to ZO-1 in the cysts, some MST3 was still present in the cytosol and basal site of the cysts at Day 8 of culture. Take a closer look in white solid-line boxed areas, the majority of kinase-dead MST3 were localized at subapical close to apical ZO1, at Days 4 and 8 of culture (Fig. 6). These results indicate that MST3 might have to be deactivated during cyst formation and then MST3 could be delivered to subapical sites.

Figure legend:

The white solid-line boxed areas were enlarged and shown below. The arrows indicated HA tagged proteins (green) localization. The HA-MST3 were away from ZO1 (red), localized at the cytosol; HA-MST3 KD were close to ZO1, localized at subapical.

---

## [Decision Letter · Decision Letter 1]

18 Apr 2023

Epithelial polarization in the 3D matrix requires MST3 signaling to regulate ZO-1 position

PONE-D-23-00666R1

Dear Dr. Lu,

We’re pleased to inform you that your manuscript has been judged scientifically suitable for publication and will be formally accepted for publication once it meets all outstanding technical requirements.

Kind regards,

Tomohito Higashi, Ph.D.

Academic Editor

PLOS ONE

Additional Editor Comments (optional):

Reviewers' comments:

Reviewer's Responses to Questions

**Comments to the Author**

1. If the authors have adequately addressed your comments raised in a previous round of review and you feel that this manuscript is now acceptable for publication, you may indicate that here to bypass the “Comments to the Author” section, enter your conflict of interest statement in the “Confidential to Editor” section, and submit your "Accept" recommendation.

Reviewer #1: All comments have been addressed

Reviewer #2: All comments have been addressed

2. Is the manuscript technically sound, and do the data support the conclusions?

Reviewer #1: Yes

Reviewer #2: Yes

3. Has the statistical analysis been performed appropriately and rigorously? 

Reviewer #1: Yes

Reviewer #2: I Don't Know

4. Have the authors made all data underlying the findings in their manuscript fully available?

Reviewer #1: Yes

Reviewer #2: Yes

5. Is the manuscript presented in an intelligible fashion and written in standard English?

Reviewer #1: Yes

Reviewer #2: Yes

6. Review Comments to the Author

Reviewer #1: The authors have addressed all of the comments in a satisfactory manner. No further comments. Recommend to accept for publication.

Reviewer #2: The author somewhat addressed my comments, the data quality is improved from previous version. The author cited a couple of references to support the appropriate cell polarization for culture on coverslips with collagen coating. The quick and easy proof should be supplying high-resolution images with z-axis information.

7. PLOS authors have the option to publish the peer review history of their article (what does this mean?). If published, this will include your full peer review and any attached files.

Reviewer #1: No

Reviewer #2: No

---

## [Editor Report · Acceptance letter]

25 Apr 2023

PONE-D-23-00666R1 

*Epithelial polarization in the 3D matrix requires MST3 signaling to regulate ZO-1 position*

Dear Dr. Lu:

I'm pleased to inform you that your manuscript has been deemed suitable for publication in PLOS ONE. Congratulations! Your manuscript is now with our production department. 

Kind regards, 

on behalf of

Dr. Tomohito Higashi 

Academic Editor

PLOS ONE